# Hybrid Immunity Results in Enhanced and More Sustained Antibody Responses after the Second Sinovac-CoronaVac Dose in a Brazilian Cohort: DETECTCoV-19 Cohort

**DOI:** 10.3390/v15101987

**Published:** 2023-09-23

**Authors:** Bárbara Batista Salgado, Aguyda Rayany Cavalcante Barbosa, Ana Ruth Arcanjo, Daniel Barros de Castro, Tatyana Costa Amorim Ramos, Felipe Naveca, Daniel M. Altmann, Rosemary J. Boyton, Jaila Dias Borges Lalwani, Pritesh Lalwani

**Affiliations:** 1Instituto Leônidas e Maria Deane (ILMD), Fiocruz Amazônia, Rua Terezina, 476 Adrianópolis, Manaus 69057-070, AM, Brazil; barbara.salgado@fiocruz.br (B.B.S.); aguyda.barbosa@fiocruz.br (A.R.C.B.); felipe.naveca@fiocruz.br (F.N.); 2Laboratory of Infectious Diseases and Immunology, ILMD/Fiocruz Amazônia and PPGIBA/ICB-UFAM, Manaus 69080-900, AM, Brazil; jaila@ufam.edu.br; 3Fundação de Vigilância em Saúde do Amazonas (FVS/AM), Manaus 69093-018, AM, Brazil; anarcanjo@fmt.am.gov.br (A.R.A.); ditec@fvs.am.gov.br (D.B.d.C.); tcaramos@hotmail.com (T.C.A.R.); 4Department of Immunology and Inflammation, Imperial College London, London W12 0NN, UK; d.altmann@imperial.ac.uk; 5Lung Division, Royal Brompton and Harefield Hospitals, Guy’s and St Thomas’ NHS Foundation Trust, London SW3 6LY, UK; r.boyton@imperial.ac.uk; 6Department of Infectious Disease, Imperial College London, London W12 0NN, UK; 7Faculdade de Ciências Farmacêuticas (FCF), Universidade Federal do Amazonas (UFAM), Manaus 69080-900, AM, Brazil

**Keywords:** vaccine, CoronaVac, breakthrough, SARS-CoV-2, antibody response

## Abstract

We measured anti-SARS-CoV-2 antibody responses before and after CoronaVac (inactivated) vaccination in a case–control study performed in CoronaVac-immunized individuals participating in a longitudinal prospective study of adults in Manaus (DETECTCoV-19). Antibody responses were measured by standard serological immunoassays. Peak anti-S-RBD and neutralizing RBD-ACE2 blocking antibody responses after two doses of CoronaVac vaccine were similar in vaccine breakthrough cases (*n* = 9) and matched controls (*n* = 45). Individuals with hybrid immunity resulting from prior SARS-CoV-2 infection followed by vaccination (*n* = 22) had elevated levels of anti-N, anti-S-RBD and RBD-ACE2 blocking antibodies after the second vaccine dose compared to infection-naïve individuals (*n* = 48). Post-vaccination SARS-CoV-2-specific antibody responses rapidly waned in infection-naïve individuals. Antibody responses wane after vaccination, making individuals susceptible to infection by SARS-CoV-2 variants. These findings support the need for booster doses after primary vaccination. Population antibody serosurveys provide critical information toward implementing optimal timing of booster doses.

## 1. Introduction

CoronaVac is a cell culture-based inactivated whole virus (SARS-CoV-2 CZ02 strain) vaccine, based on the principle of formaldehyde and β-propiolactone inactivation and addition of aluminum-hydroxide as a vaccine adjuvant. Since availability of COVID-19 vaccines, 47 countries worldwide have used Sinovac-CoronaVac vaccine to vaccinate frontline health workers and the general population against SARS-CoV-2. In December 2022, the huge wave of infections in China and neighboring countries has raised concern about the durability of immune response after primary series of two doses and the need for booster doses. CoronaVac was shown to increase anti-S-RBD antibody responses post-vaccination and vaccine effectiveness toward hospitalization was 87.5% (95% CI, 86.7–88.2%) during the initial phase of the pandemic. In Chile, after two doses vaccine effectiveness in preventing hospitalization and ICU admission was reduced in adults aged ≥60 years as compared to that in adults aged 16–59 years. In Turkey, the hospitalization rate was similar in the breakthrough cases and those who had COVID-19 before CoronaVac vaccination [1]. On the other hand, breakthrough infection by alpha, beta, delta [2], and omicron [3] variants are described. Low levels of neutralizing antibodies against gamma variant were detected in CoronaVac vaccinees [4]. In October 2021, WHO recommended an additional dose for older adults and immunocompromised persons who have received two doses of CoronaVac to ensure sufficient protection. However, the rate of administration and uptake of booster doses has been slow in most countries compared to primary vaccine doses. Also, circulation of the more transmissible omicron subvariants has had a significant impact on frontline workers, high-risk populations, and unvaccinated individuals. Most vaccine response studies have focused on frontline healthcare workers. In this study, we looked at COVID-19 vaccine response in an adult population cohort (DETECTCoV-19) in Manaus [5,6]. Here, we performed a detailed analysis of anti-SARS-CoV-2 antibody responses among CoronaVac vaccine breakthrough cases compared to uninfected controls and performed a longitudinal evaluation before and after vaccination to document waning of antibody responses among individuals with or without previous COVID-19.

## 2. Methods

### 2.1. Study Setting

The longitudinal study, DETECTCoV-19, in Manaus was designed to follow-up ≥18-year-old adults initially recruited between 19 August 2020 and 2 October 2020 [5,6]. The objective of this cohort is to understand the epidemiology and immune response toward SARS-CoV-2 in Manaus city.

Study participants were tested longitudinally for anti-N and anti-S-RBD SARS-CoV-2 antibodies. RT-PCR testing for SARS-CoV-2 was available for symptomatic individuals and contacts. Since June 2021, we have offered SARS-CoV-2 antigen-detecting rapid lateral flow tests (Ag-RDT, Siemens Healthineers, Erlangen, Germany) in addition to RT-PCR testing for symptomatic and asymptomatic individuals. At every visit, an interviewer collected information related to COVID-19 disease symptoms or contact, social distancing and protective practices, self-medication and prescribed medication used for treatment of symptoms or prevention of COVID-19 as well as COVID-19 vaccination history. Currently in Manaus, depending on the availability, adults are vaccinated with Sinovac-CoronaVac, Oxford-AstraZeneca, Pfizer-BioNtech or Janssen COVID-19 vaccines. The research ethics committee of Federal University of Amazonas (UFAM) approved this study (CAAE:34906920.4.0000.5020) in accordance with Brazilian law, and the Helsinki declaration. All the participants gave oral and written consent prior to enrolment.

### 2.2. Antigen POC Test, RT-PCR, and SARS-CoV-2 Variant Identification

Nasopharyngeal swabs were collected by trained personnel. All symptomatic individuals were tested by SARS-CoV-2 antigen-detecting rapid lateral flow POC tests (Ag-RDT, Siemens Healthineers) followed by RT-PCR test (Biomanginhos, Rio de Janeiro, Brazil). We offered Ag-RDT testing for asymptomatic individuals during the study visits. Ag-RDT positive samples were tested by RT-PCR and findings expressed as the cycle threshold (Ct) for the gene encoding the nucleocapsid protein (N gene). Whole genome sequencing was performed to identify variant of concern (VOC) with the use of COVIDSeq library preparation kit (Illumina, San Diego, CA, USA) for samples with Ct value equal or below thirty, as previously described [7].

### 2.3. Case–Control Study Design

A breakthrough vaccine infection was defined as the detection of SARS-CoV-2 by RT-PCR performed 15 or more days after receipt of second dose of the vaccine. Between June and November 2021, for the case–control analysis we selected for each breakthrough case, at-least five matched controls from the pool of cohort participants, according to the following variables: sex, age (+/−2 years), COVID-19 vaccine, COVID-19 vaccine interval between the two doses and previous COVID-19 history. When more than five controls were available for each case, we randomly selected five controls out of the available options. We performed a second algorithm where cases and controls were paired additionally with variable co-morbidity. No patient had immunosuppression.

### 2.4. Serological Assays

Longitudinal plasma samples collected before and after vaccination for cases and controls were tested by three different assays to assess the humoral response. An indirect enzyme-linked immunosorbent assay (ELISA)-based serological assay was used to measure anti-SARS-CoV-2 nucleocapsid (N) IgG antibodies in plasma samples using recombinant full-length SARS-CoV-2 N protein, as previously described [5,6]. All samples with a reactivity index (RI) value above 1.5 (assay cut-off) were considered positive for anti-N IgG antibodies.

Anti-S-RBD IgG antibodies were detected using the QuantiVac assay (Euroimmun, Lübeck, Germany) as per manufacturer’s instructions and antibody titers were expressed as binding antibody units (BAU)/mL; samples above 35.2 BAU/mL were considered positive as per manufacturer’s instructions.

NeutraLISA (Euroimmun), a commercial competitive assay, was used as per manufacturer’s instructions to estimate potentially neutralizing antibodies inhibiting the binding of SARS-CoV-2 spike protein receptor-binding domain (RBD) to ACE2 receptors of the host cells. Assay results were expressed in percentage inhibition (IH) of RBD-ACE2 interaction: %IH = 100% − [Optical density of patient sample × 100%/optical density of blank (mean)]. Upper threshold of the normal range (cut-off value) was set at 25% IH as per the manufacturer’s instructions.

### 2.5. Statistical Analysis

Chi-square tests or Fisher’s exact for two-by-two contingency tables were used to examine the statistical significance and association between study variables. Paired and unpaired *t*-tests were performed to understand the differences in the antibody titers between the two study groups. Paired *t*-tests or one-way ANOVA tests with Tukey’s post hoc test were used. All statistical analysis were performed using the GraphPad Prism software (v9.1.2 for Mac OS).

## 3. Results

### 3.1. Vaccination and Identification of Breakthrough Infections

Manaus started vaccinating healthcare professionals in January 2021 with Sinovac-CoronaVac vaccine and subsequently, the general population with Sinovac-CoronaVac, Oxford-AstraZeneca, Pfizer-BioNTech and Janssen vaccine depending on the availability of the vaccine doses. SARS-CoV-2 testing by RT-PCR and rapid COVID-19 antigen test for symptomatic and asymptomatic infections identified 4.8% (21/429) and 4.18% (18/430) positive for SARS-CoV-2 infection by RT-PCR and rapid COVID-19 antigen test between June and October 2021, respectively. Vaccine breakthrough was defined as RT-PCR confirmed SARS-CoV-2 infection 15 days after the second vaccine dose (Sinovac-CoronaVac, Oxford-AstraZeneca and Pfizer-BioNTech). In total, we identified ten breakthrough infections after full-vaccination, eight individuals with infection after first vaccine dose and three infections among unvaccinated individuals. Of the ten vaccine breakthrough cases, nine received Sinovac-CoronaVac and one received Oxford-AstraZeneca vaccine. Illumina sequencing confirmed 90% (9/10) of individuals were infected with the gamma SARS-CoV-2 VOC, one patient had RT-PCR Ct value >30 and was not sequenced.

Of the Sinovac-CoronaVac vaccine breakthrough patients, 56% (5/9) were female, with a median age of 38 years (IQR, 28.5–54 years), and 67% (6/9) identified as having a comorbidity (Table 1). A median of 51 days (IQR 49–78 days) after complete vaccination, SARS-CoV-2 breakthrough infection was confirmed by RT-PCR and/or antigen test. Among the nine CoronaVac patients, one individual had asymptomatic SARS-CoV-2 infection, two required hospitalization and one required oxygen support (Table 1). Mild symptoms were observed after 21 days of the second vaccine dose in the Oxford-AstraZeneca vaccine breakthrough (a 41-year-old woman).

We did not perform epidemiologic investigation or contact tracing. All symptomatic individuals were advised to follow isolation protocols to limit virus transmission. In 43% (3/7 cases) of individuals for whom data were available about the source of infection, the suspected source was an unvaccinated household member. Three individuals lived alone and did not know the infection source.

### 3.2. CoronaVac Breakthrough Case and Uninfected Control Analysis

Table 1 compares population characteristics and disease outcomes between CoronaVac vaccine breakthrough cases (*n* = 9) and CoronaVac uninfected matched controls (*n* = 45). For every CoronaVac breakthrough case we selected five random controls as described in the Methods section (Figure 1A). We did not have blood samples just before the breakthrough infection, hence, humoral responses among cases and controls were compared at baseline before vaccination (median case 20 days; median control 22 days) and peak response after second vaccine dose (median case, 58 days; median control, 42 days). At baseline before vaccination, 22% of cases and controls were positive for anti-SARS-CoV-2 N protein IgG antibodies; upon Sinovac-CoronaVac vaccination, 75% and 78% of cases and controls had detectable anti-N IgG antibodies, respectively. Upon receiving two doses of Sinovac-CoronaVac, 100% of cases and 91% of controls had detectable anti-S-RBD IgG antibodies. Post-vaccination, 70% of cases and 47% of controls had detectable RBD-ACE2 inhibitory antibodies above the cut-off limit. We did not observe a statistically significant difference in the anti-N and anti-S antibody titers between cases and controls (Figure 1B,C).

In a secondary analysis, we included other uninfected matched controls (*n* = 61) paired only by age, sex, and previous COVID-19 history, and compared their humoral responses with the breakthrough cases (Appendix A). In general, we did not observe statistically significant differences between cases and controls in terms of percentage positivity and antibody titers for anti-nucleocapsid (N) IgG, anti-spike (S)-receptor binding domain (RBD) IgG and RBD-ACE2 inhibitory antibodies. Longitudinal data for each breakthrough case are depicted in Appendix A.

### 3.3. CoronaVac Vaccine Response among Naïve and Previous COVID-19

To understand the underlying differences in the immune response among individuals with complete vaccination, we compared infection-naïve, and SARS-CoV-2-infected individuals before vaccination. Firstly, before vaccination individuals with previous COVID-19 had significantly higher antibody titers toward N, S-RBD and RBD-ACE2 inhibitory antibodies compared to naïve individuals (Figure 2). Secondly, we observed 100% seroconversion after two vaccine doses for N and S protein among the COVID-19 positive individuals compared to the COVID-19-naïve group (Appendix A). Also, anti-N and anti-S-RBD geometric mean antibody titers were significantly higher among the SARS-CoV-2 infected group in comparison to the naïve group. We additionally observed that SARS-CoV-2-infected individuals had enhanced plasma antibody titers post vaccination (Figure 3) including increased RBD-ACE2 inhibitory antibodies. In a longitudinal comparison we observed a more rapid decay in anti-N, anti-S-RBD and RBD-ACE2 inhibitory antibodies titers among the naïve group compared to the previously SARS-CoV-2-infected individuals (Figure 3).

We also compared Sinovac-CoronaVac and Oxford-AstraZeneca COVID-19 vaccine responses. Oxford-AstraZeneca vaccine was initially only available to over sixty-year-old individuals in Manaus. Since we had a small pool of completely vaccinated Oxford-AstraZeneca individuals, we could not perform a stringent matching as performed for the CoronaVac case–control analysis. Also, the recommended duration between two Oxford-AstraZeneca vaccines is three months, compared to 28 days for the Sinovac-CoronaVac vaccine. Therefore, we randomly selected 66 Oxford-AstraZeneca vaccinees and compared their immune responses to people receiving Sinovac-CoronaVac vaccine. Appendix A details the population characteristics of the two vaccine groups. A median of 53 days after the second dose was noted for both the vaccine groups. Overall, anti-S-RBD IgG antibody titers and RBD-ACE2 inhibitory antibody levels were significantly higher among Oxford-AstraZeneca vaccinees. We observed a significant increase in anti-spike-RBD titers and RBD-ACE2 inhibitory antibodies among COVID-19-naïve individuals vaccinated with Oxford-AstraZeneca compared to Sinovac-CoronaVac vaccine. We also observed elevated humoral response in previously COVID-19-infected individuals compared to naïve individuals independent of the vaccine platform. Oxford-AstraZeneca was a better immunogen compared to the Sinovac-CoronaVac as observed in the serological assays (Appendix A).

In summary, previous SARS-CoV-2 infection boosted neutralizing antibodies post vaccination. We observed a rapid decay in anti-N and anti-S-RBD antibodies among naïve individuals compared to individuals with previous SARS-CoV-2 infection.

## 4. Discussion

Our study examined antibody waning and vaccine breakthrough infection after the second CoronaVac vaccine dose. Anti-S-RBD levels peaked three to four weeks following the second dose of vaccine and the geometric mean antibody titers were higher among individuals with COVID-19 infection before vaccination. There was substantial waning of anti-S-RBD following vaccination that declined following a log-linear course. In a longitudinal comparison of CoronaVac breakthrough cases and uninfected controls, peak neutralizing antibody response was similar after vaccination and antibody titers were not associated with the risk of a breakthrough infection. During our study period, gamma—followed by omicron—was the dominant infecting strain in Manaus, Amazonas, Brazil. In this analysis, we also observed that COVID-19-naïve vaccinees had increased post second dose vaccine antibody responses compared to those with previous COVID-19.

Strengths of our analysis include longitudinal analysis of serological response before and after vaccination, and case–control controlled analysis (for variables such as age, sex, SARS-CoV-2 infection history and comorbidities). Voluntary RT-PCR testing identified 4.89% SARS-CoV-2 positive individuals during the study period and a low rate of gamma variant breakthrough infection (0.8%, 10/1239) among fully vaccinated individuals. CoronaVac vaccine breakthrough was 1% (9/890); however, we observed a higher rate of breakthrough infections among naïve individuals (2%, 7/350) compared to individuals with previous COVID-19 (0.4%, 2/540) infection. Overall, hybrid immunity induced by natural infection and vaccination could provide robust protection against breakthrough infection. Previous studies in Chile [8] and Turkey [9] reported higher breakthrough infection; on the other hand, CoronaVac vaccine was shown to reduce hospitalization after full vaccination. In Brazil, the protection against SARS-CoV-2 gamma variant infection was 50% after CoronaVac vaccination among frontline healthcare workers [10].

Vaccine breakthrough cases and matched controls had similar levels of anti-S-RBD geometric mean antibody titers. The observed breakthrough infection among vaccinated individuals could have been a result of noncompliance of non-pharmaceutical intervention measures such as not wearing a mask during contact. Relaxation of protective measures was associated with the risk of SARS-CoV-2 [6]. However, SARS-CoV-2 infection before vaccination contributed to a slower antibody decay rate and elevated geometric mean antibody titers. Thus, the rate of decay of anti-N and anti-S-RBD antibody levels was slower compared to the COVID-naïve individuals. Additionally, we observed a significantly elevated geometric mean in anti-S titers and neutralizing antibody response after the second dose of Oxford-AstraZeneca, an adenovirus vaccine, compared to CoronaVac, an inactivated vaccine.

Worldwide, omicron has replaced other SARS-CoV-2 variants, and overwhelming data not only confirm the low vaccine coverage in most low-income countries [11], but also decreased effectiveness of mRNA, adenovirus vectored and inactivated vaccines against emerging subvariants [12,13]. Omicron variants have been linked to deaths among unvaccinated and elderly individuals with poor immune response. Hence, most developed countries have administered additional COVID-19 doses to boost waning immune response. Bivalent mRNA COVID-19 vaccines containing the original SARS-CoV-2 variants and omicron subvariants BA.1 and BA.4/5 have been available in few select countries. Overall, we still do not know the impact of the omicron subvariants in the unvaccinated individuals or countries with low vaccine coverage and those vaccinated with low immunogenic vaccines [14]. Hence, measuring spike-RBD binding antibodies in the general population can be used as surrogate for deciding on booster doses and determining post-vaccination population-level immunity against the ancestral strain.

During the first two years of the SARS-CoV-2 pandemic, several variants were reported from different countries; however, over the last year, omicron subvariants have dominated coronavirus circulation. We do not know much about the evolution of the coronaviruses, and this will be a challenge in understanding the correlates of protection for SARS-CoV-2 to create universal vaccine design against SARS-CoV-2 variants [15]. However, vaccines that improve T-cell response have been shown to be effective in reducing hospitalization and mortality among the vaccinated. Also, heterologous boosting with adenovirus vectored or mRNA COVID-19 vaccines might be a way forward to improve durability and persistence of humoral and cellular immune response [16,17]. Since COVID-19 vaccines vary regarding immunogenicity and immune specificity, a vaccine booster dose with a T-cell–skewed vaccine will not only boost T-cell responses but also potentially reduce the chance of breakthrough infection through the generation of cross-protective T-cell immunity [18].

Limitations of this study include the small sample size of breakthrough infection and the focus on voluntary testing to identify SARS-CoV-2 infections. Asymptomatic infections were not tested for and could, therefore, be missed in the cohort chosen as control, which in turn may cause a misinterpretation of the results regarding the comparison with the immune response elicited by the breakthrough cases. Therefore, our conclusions are directed toward symptomatic infection. It will be important to investigate whether the duration of cross-reactive antibody in breakthrough infections is more prolonged than following booster vaccination. Patients were not matched for their behavioral and social changes. Thus, the risk of acquiring might be different for the control and case groups. We did not measure T-cell response in this study; recent studies have demonstrated the importance of T-cell response in protecting against COVID-19 severe disease. Hybrid-immunity derived from virus infection and vaccination could potentially boost both humoral and cellular response, providing improved protection in individuals compared to individuals with only two vaccine doses.

CoronaVac, a COVID-19 vaccine, has been used among frontline healthcare providers in Asia, Africa, South America, North America and Oceania and ranks as the vaccine with most doses administered worldwide. Observed waning neutralizing antibody response post vaccination makes individuals susceptible to higher transmissible and immune escape SARS-CoV-2 variants, hence a booster dose after primary vaccination of CoronaVac is required to heighten immune response. With the aim of reducing severe disease among high-risk patients, a COVID-19 vaccine will probably be included in the yearly vaccination calendar together with the influenza vaccine. Mask use has been declining in most countries recently, and the recent outbreaks after lifting COVID-19 restrictions could impact SARS-CoV-2 spillover in other countries with low vaccination coverage and seronegative COVID-19-vaccinated high-risk populations. Hence, compliance with non-pharmaceutical intervention measures, such as wearing a mask, by symptomatic individuals can significantly reduce transmission of respiratory viruses to the general public.

## Figures and Tables

**Figure 1 viruses-15-01987-f001:**
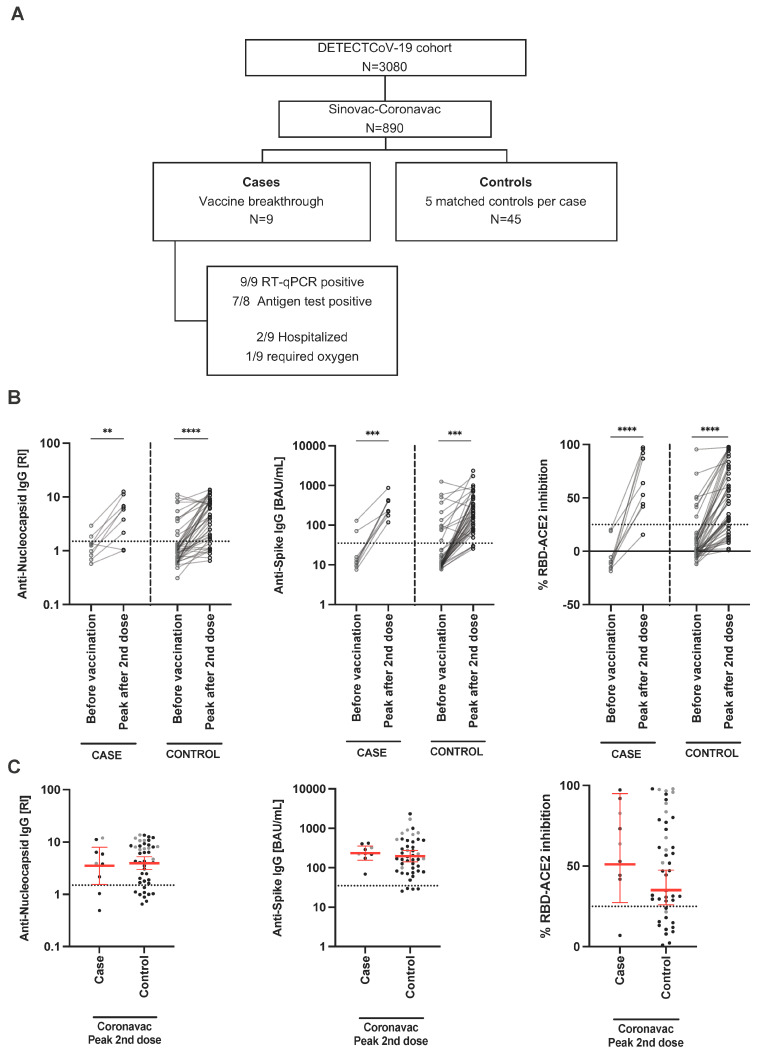
Equivalent humoral response between Sinovac-CoronaVac vaccine breakthrough cases and controls. (**A**) Symptomatic vaccine breakthrough cases were paired with age, sex, time of vaccination and previous COVID-19 history to vaccinated controls without breakthrough infection during the same period. Immunoassays measured serum anti-nucleocapsid IgG, anti-spike-RBD IgG and %RBD-ACE2 inhibitory antibodies before and after immunization. (**B**) Paired comparison of humoral response before and after vaccination. (**C**) Peak humoral response after two doses. Median is represented by red horizontal lines. Horizontal dotted lines represent assay cut-off. Each patient represents one data point on the graph. Paired or unpaired *t*-test was performed to compare patient groups. ** *p* < 0.01, *** *p* < 0.001, **** *p* < 0.0001.

**Figure 2 viruses-15-01987-f002:**
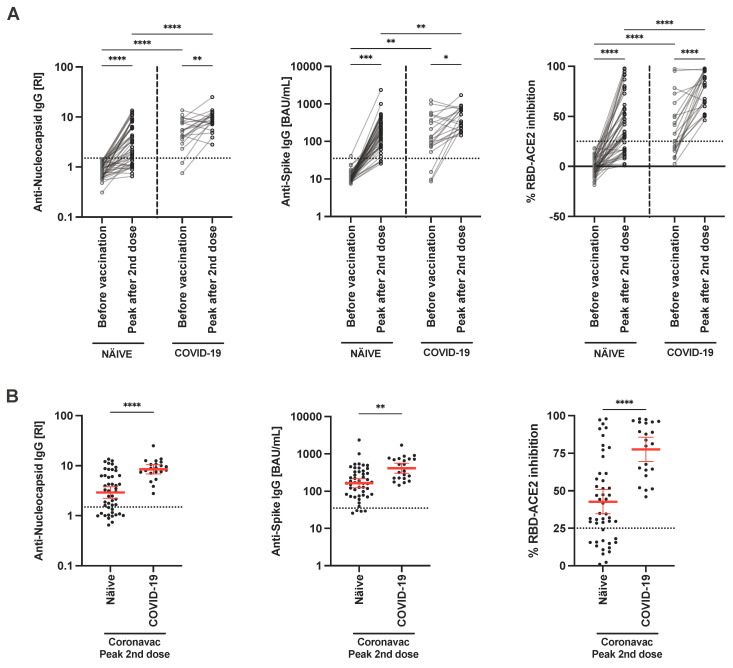
Elevated humoral response after Sinovac-CoronaVac vaccination among COVID-19 infected individuals. Immunoassays measured serum anti-nucleocapsid IgG, anti-spike-RBD IgG and %RBD-ACE2 inhibitory antibodies before and after immunization. (**A**) Paired comparison of humoral response among naïve and infected individuals before and after two doses of inactivated COVID-19 vaccination. (**B**) Peak humoral response after two doses. Median is represented by red horizontal lines. Horizontal dotted lines represent assay cut-off. Each patient represents one data point on the graph. Paired or unpaired *t*-test was performed to compare patient groups. * *p* < 0.05, ** *p* < 0.01, *** *p* < 0.001, **** *p* < 0.0001.

**Figure 3 viruses-15-01987-f003:**
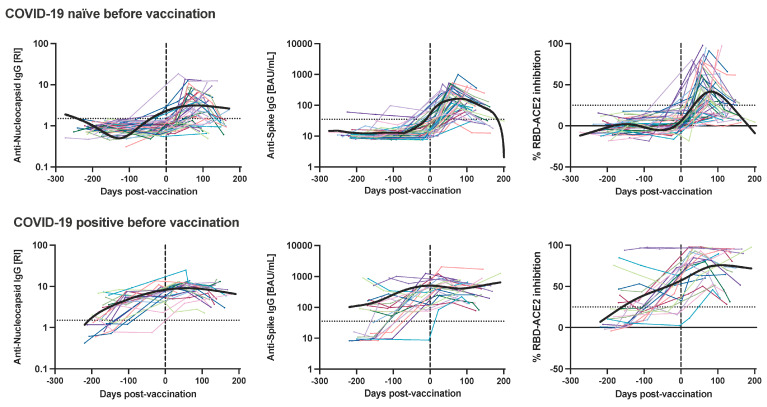
Hybrid immunity results in sustained antibody responses after the second Sinovac-CoronaVac dose. Longitudinal serum anti-nucleocapsid IgG, anti-spike-RBD IgG and %RBD-ACE2 inhibitory antibodies among naïve and infected individuals before and after two doses of inactivated COVID-19 vaccination. Each patient is represented by a colored line. Bold black line depicts the trend. Splines were plotted using the Fit Spline program in GraphPad Prism software.

**Table 1 viruses-15-01987-t001:** Populational characteristics, SARS-CoV-2 infection and seroreactivity.

		Case (*n* = 9)	Control (*n* = 45)	*p* Value
Characteristics				
Female, n (%)		5 (55.6)	25 (55.6)	>0.9999 *
Age, mean		40.2	40.6	0.9399 ^‡^
Age, median (IQR)		38 (28.5–54)	36 (30–54)	
Age, range		24–67	22–69	
Income	0–3 minimum salaries	3 (33.3)	7 (15.6)	0.4403 *
	4–6 minimum salaries	2 (22.2)	15 (33.3)	
	>6 minimum salaries	4 (44.4)	23 (51.1)	
Comorbidities, yes, n (%)		6 (66.7)	28 (62.2)	0.801 *
Diabetes		1 (11.1)	5 (11.1)	
Hypertension		3 (33.3)	15 (33.3)	
Obesity		3 (33.3)	9 (20)	
Asthma		1 (11.1)	4 (8.9)	
Cardiopathy or nephropathy		1 (11.1)	1 (2.2)	
Interval from first to second dose of vaccination, days median (IQR)		28.0 (24.0–30.5)	25.0 (23.0–35.0)	0.0565 ^‡^
COVID-19-related information				
COVID-19 prior to vaccination, yes, n (%)		2 (22.2)	10 (22.2)	>0.9999 *
Days of onset symptoms after full vaccination ^$^, median (IQR)		61 (49.3–78)		
Days with symptoms on day of antigen testing after fully vaccinated, median (IQR)		4 (2–6)		
Antigen test after fully vaccinated ^$^		7/9 positive		
Days with symptoms on day of RT-qPCR testing after fully vaccinated, median (IQR)		4 (2–6)		
RT-qPCR testing after fully vaccinated ^$^		9/9 positive		
Hospitalization, yes, n (%)		2 (22.2)		
Oxygen requirement, yes, n (%)		1 (11.1)		
Symptoms on the day of testing, yes, n (%)		8 (88.9)		
Symptoms, n (%)	Fever	6 (75.0)		
	Dry cough	6 (75.0)		
	Sore throat	3 (37.5)		
	Nasal discharge	7 (87.5)		
	Chill	3 (37.5)		
	Headache	7 (87.5)		
	Myalgia	5 (62.5)		
	Arthralgia	1 (12.5)		
	Fatigue	3 (37.5)		
	Chest pain	2 (25.0)		
	Backache	5 (62.5)		
	Shortness of breath	1 (12.5)		
	Anosmia	3 (37.5)		
	Dysgeusia (loss of taste)	3 (37.5)		
	Conjunctivitis	2 (25.0)		
	Loss of appetite	1 (12.5)		
Serological testing				
Before vaccination	Anti-nucleocapsid IgG positive, n (%)	2 (22.2)	10 (22.2)	>0.9999 ^‡^
	Anti-nucleocapsid IgG (RI), median (IQR)	1.25 (0.75–1.6)	1.02 (0.65–1.35)	0.4391 ^‡^
	Anti-spike-RBD IgG positive, n (%)	2 (22.2)	11 (24.4)	0.8868 ^‡^
	Anti-spike-RBD IgG (BAU/mL), median (IQR)	11.78 (9.4–43.35)	10.86 (9.5–32.5)	0.4363 ^‡^
	RBD-ACE2 inhibition (%), median (95% CI)	−9.99 (−16.3–18.9)	1.19 (−1.1–7.0)	0.0627 ^‡^
Peak response after fully vaccinated ^$^	Anti-nucleocapsid IgG positive, n (%)	6 (66.7)	35 (77.8)	0.4766 ^‡^
	Anti-nucleocapsid IgG (RI), median (IQR)	5.88 (1.6–8.9)	4.48 (1.7–8.5)	0.9143 ^‡^
	Anti-spike-RBD IgG positive, n (%)	9 (100.0)	41 (91.1)	0.3526 ^‡^
	Anti-spike-RBD IgG (BAU/mL), median (IQR)	228.8 (197.3–422.4)	189.9 (82.5–496.0)	0.8714 ^‡^
	RBD-ACE2 inhibition (%), median (95% CI)	63.8 (41.8–95.7)	46.9 (30.8–61.5)	0.1515 ^‡^

^$^ 15 days after second dose of vaccine was considered fully vaccinated. RI: Reactivity Index. BAU: binding antibody units. IQR: interquartile. IgG: immunoglobulin G. RBD: receptor binding domain. ACE2: angiotensin-converting enzyme-2. ^‡^ Unpaired *t*-test. * Chi-square test.

## Data Availability

De-identified participant data can be made available to researchers after approval from the research ethics committee. The requests should be directed to the corresponding author (pritesh.lalwani@fiocruz.br).

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
