# Peer review of "Hybrid Immunity Results in Enhanced and More Sustained Antibody Responses after the Second Sinovac-CoronaVac Dose in a Brazilian Cohort: DETECTCoV-19 Cohort"

_viruses, 2023, doi:10.3390/v15101987_

Round 1

Reviewer 1 Report

Paper presents intersting result on Hybrid immunity results in enhanced and more sustained antibody responses after the second Sinovac-Coronavac dose in a 3 Brazilian cohort: DETECTCoV-19 cohort.

The strenght of the paper is the longitudinal analysis. However it could be interestiong to analyze differences with previopsu report on breakthroughafter different vaccine schedule.

This is not mandatory but would be interesting to add to the paper

Minor spell check are required

Tables shoudl be improved

references should be done according to journal rules

Few mnor spell check

Author Response

We thank Reviewer 1 for the comments. We have made the necessary corrections. 

Reviewer 2 Report

  The authors measured anti-SARS-CoV-2 antibodies before and after vaccination with SinoVac-CoronaVac COVID-19 vaccine. They found peak neutralizing antibody response after vaccination was similar in vaccine breakthrough cases and matched controls, suggesting antibody titers were not associated with the risk of a breakthrough infection. They also showed individuals with hybrid immunity resulting from prior SARS-CoV-2 infection followed by vaccination had elevated levels and a slower decay rate of anti-SARS-CoV-2 antibodies compared to COVID-19 naïve individuals after vaccination. Since the antibody response post vaccination rapidly waned in the naïve individuals, the authors advocated the need for booster doses after primary vaccination of CoronaVac. The major concern the reviewer has is the small sample size in this study. However, the manuscript is well-organized and includes important information. I have raised several points which need to be clarified. These are given below.

Specific points:

1.     In this study, it was suggested antibody titers post vaccination were not associated with the risk of a breakthrough infection. How do the authors think of cellular immunity post vaccination in the risk of a breakthrough infection? 

2.     Is there any difference in the rate of a breakthrough infection between individuals with hybrid immunity resulting from prior SARS-CoV-2 infection followed by vaccination and COVID-19 naïve individuals with vaccination?

3.     Minor points:

(1)   Figure legend of Fig. 2: (B)  (A)?; (C)  (B)?  

(2)   Page 8, line 237: “Supplementary table 3” should be “Supplementary Figure 3”?

Author Response

We thank Reviewer 2 for these important comments.

Specific points:

  1. In this study, it was suggested antibody titers post vaccination were not associated with the risk of a breakthrough infection. How do the authors think of cellular immunity post vaccination in the risk of a breakthrough infection? We do believe and literature supports an important role of T-cell response in protection against severe COVID-19 disease. In this study, we couldn't measure the T-cell response. We have included a sentence to talk about the importance of T-cell response in hybrid immunity and its role in protection. We have included this as a study limitation, please refer line 331-333.
  2. Is there any difference in the rate of a breakthrough infection between individuals with hybrid immunity resulting from prior SARS-CoV-2 infection followed by vaccination and COVID-19 naïve individuals with vaccination? We have included this important result now in our discussion, please refer line 274-278. 
  3. Minor points:

(1)   Figure legend of Fig. 2: (B) → (A)?; (C) → (B)?  We have now corrected, please refer the said figure legend.

(2)   Page 8, line 237: “Supplementary table 3” should be “Supplementary Figure 3”? We have now corrected, please read as Supplementary table 2.